# Linguistic Image Understanding

## Abstract

We present a novel, text-centric vision-language framework called Linguistic Image Understanding (LIU). It introduces a unique pipeline for image-text processing by transforming images into comprehensive textual descriptions that encapsulate not only comprehensive object semantic details but also the spatial positioning of objects within images, enriching visual grounding ability. Then LIU feeds these descriptions into pretrained large language models to handle vision-language tasks without seeing the image and achieves promising performance on many vision-language tasks with high computational efficiency and enhanced interpretability. Experimental results show that LIU exhibits a unique potential to refine and elevate the quality of existing vision-language pre-train datasets, resulting in significantly improved Image-Text Matching scores. Accordingly, vision-language models fine-tuned on these refined datasets have also shown performance improvement across a broad spectrum of vision-language tasks. Our work points to a promising future where the amalgamation of advanced language models and semantic-rich textual descriptions can drive the evolution of more efficient and adaptable vision-language models.

## 1 Introduction

Autoregressive language models that generate sequences by predicting subsequent words from prior context have seen a surge in effectiveness and popularity (Devlin et al., 2019; Touvron et al., 2023; Zhang et al., 2022; Chung et al., 2022; OpenAI, 2023b). This is attributed to the significant increase of the model parameters, from hundreds of millions (Devlin et al., 2019) to tens of billions (Touvron et al., 2023), and the availability of much larger training corpora. These advancements have largely enhanced their reasoning ability across various kinds of tasks. For instance, GPT-4 (OpenAI, 2023b) achieved a top 10% score on a simulated bar exam. Yet, how to harness the knowledge encapsulated within these Large Language Models (LLMs) to enhance vision-language tasks is in its infancy and remains many exciting and valuable research topics.

In an effort to integrate LLMs, conventional vision-language architectures predominantly utilize tons of image-text pairs to train a joint representation of visual and textual data. For instance, Frozen (Tsimpoukelli et al., 2021) freezes the language model, and then only trains the vision encoder end-to-end. The more recent works like Flamingo (Alayrac et al., 2022) and BLIP2 (Li et al., 2023) typically freeze both LLM and Vision Model, and bridges them by only training adapter layers. The reduction in trainable parameters is motivated by the desire for computational efficiency, GPU memory cost savings, and the prevention of catastrophic forgetting (Tsimpoukelli et al., 2021; Yang et al., 2022; Alayrac et al., 2022).

Unlike these prior approaches, in this paper, we introduce a very simple and effective solution, termed as "LIU". As shown in Figure 1, its main idea is to **directly transform an image into a text caption and then feed the vanilla and generated captions into a frozen large language model for vision-language tasks**. LIU first employs a vision semantic extractor to comprehend the context of an image, and then adopts a trainable language model,

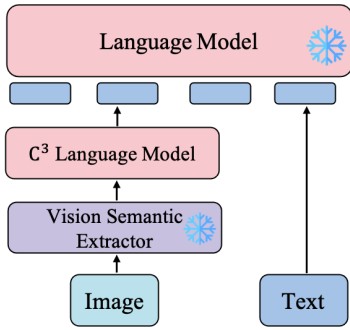

Figure 1: **Overall Framework** of LIU: we convert an image into a textual caption, and use a pretrained frozen language model to *process vision-language tasks via only the texts* in the generated and vanilla captions.

namely Comprehensive Caption Composer ($C^3$), to synthesize all semantics (e.g., objects and their positions and relations) into a comprehensive text description, namely, the image captions. Finally, LIU feeds the generated image captions and vanilla image text description into a frozen pretrained large language model, e.g. OPT (Zhang et al., 2022) and Flan-T5 (Chung et al., 2022). Our LIU enjoys several advantages which are summarized below.

**1) Efficiency:** In contrast to training previous vision-language models on vast amounts of image-text pair data, e.g., 1.8B image-text pairs for Flamingo, our LIU only needs to train $C^3$ model which is trained solely on instructional data sourced from 6,000 images without text. Once the $C^3$ model is trained, LIU can perform vision-language tasks in a zero-shot manner. Note, LIU uses the frozen vision semantic extractor to extract the semantics of all images in advance. In this way, it avoids any extra image processing during training, and thus saves large computational cost.

As an example, on the image-text retrieval task, our LIU achieves a 51.4% Recall@1 score on zero-shot COCO retrieval and only needs 5 minutes to evaluate. This performance outpaces the image-text retrieval model based on the BLIP (Li et al., 2022) system by 7.6%, even though the latter is trained for 7 hours on COCO data with the same computation resources.

**2) Interpretability.** The textual captions, generated by LIU, provide an easily comprehensible view of visual data, thereby achieving much higher interpretability than the black-box representations in adapter methods and vision-encoder training methods. More importantly, one can also easily judge whether the textual caption can well match the corresponding image by human or many existing pretrained multi-modal models, e.g. CLIP (Radford et al., 2021), GIT (Wang et al., 2022a) and Flamingo (Alayrac et al., 2022). For those improper captions, one can use or build other caption composer models, e.g. Alpaca (Taori et al., 2023) and GPT-4 (OpenAI, 2023b), to regenerate it, which should further improve the performance but is beyond the current scope of this work.

**3) More Applications.** Beyond vision-language tasks, LIU can also handle many other applications, such as dataset refinement that aims to regenerate the captions of the images to improve caption accuracy for image-text pairs data and privacy protection that regenerates the captions of the images according to certain requirements, e.g. avoiding sensitive information by removing keywords.

In this work, we are particularly interested in dataset refinement because of its border applications. As an example, we apply LIU to address significant misalignment issues prevalent in large image-text pre-training datasets, and significantly improve the Image-Text Matching (ITM) score. When further fine-tuned on this refined dataset, our model consistently delivers improved results across a wide array of vision-language tasks, as exemplified by the improvement from 121.6 to 127.2 in the CIDEr metric on the COCO Captioning task.

By leveraging the power of LLMs and the inherent semantic richness of the text, LIU opens a new pathway towards more efficient, versatile, and interpretable vision-language applications.

## 2 METHOD

### 2.1 CONCEPTUAL FOUNDATION

Language, a fundamental tool for human communication, enables us to encapsulate experiences and observations, especially the visual ones (Evans & Levinson, 2009). This cognitive ability to transform visual information into textual descriptions, as illustrated by the left figure of Figure 2, is a cornerstone of human interaction that we seek to emulate in machine learning models. We hypothesize that **the semantics of an image can be effectively represented by objects and their relative positions without requiring overly precise positional information**. This work aims to develop a model capable of abstracting an image into text (namely, an image caption), akin to the natural human process of describing visual experiences. By doing so, we strive to contribute to the evolution of more interpretable and flexible visual-language models.

### 2.2 LINGUISTIC IMAGE UNDERSTANDING

Building upon the above conceptual foundation, our objective is to directly convert an image into a textual caption and subsequently input both the original and generated captions into a pre-trained and frozen large language model for vision-language tasks. To this end, we proposed LIU frame-

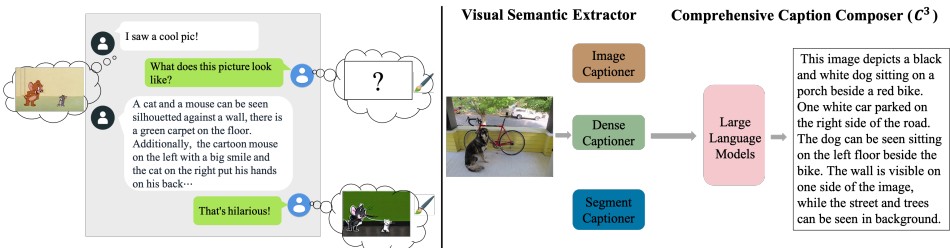

Figure 2: **Exploring the Conceptual Foundation and Core Pipeline of** LIU. Left: We posit that language serves as a critical bridge in understanding the world. Right: The two key components in our LIU. Initially, the visual semantic extractor with three components, i.e., image captioner, dense captioner and segment captioner, are employed to extract different visual semantics of an image. Then, a Large Language Model (LLM), i.e., Comprehensive Caption Composer, is trained to summarize and synthesize these visual semantics for a more comprehensive image caption.

work whose comprehensive pipeline is illustrated in Figure 1. Specifically, LIU employs a vision semantic extractor to gain a deep understanding of the image. It then utilizes a trainable language model, known as the Comprehensive Caption Composer ($C^3$), to generate a textual representation encompassing all pertinent semantics of the image, including objects and their respective positions. Finally, the generated caption, alongside the original image caption, is fed into a pre-trained and frozen large language model, such as OPT or Flan-T5, for performing vision-language tasks.

The key challenge for LIU is how to convert an image into a comprehensive and accurate text caption via the two components, namely, "visual semantic extractor" and "Comprehensive Caption Composer ($C^3$)". To this end, as shown in Figure 2 (right), LIU builds its visual semantic extractor that contains an image captioner, a dense captioner, and also a segmentation captioner to generate image captions from a coarse-grained level to a fine-grained level. In this way, LIU can provide three different granular captions. Then, LIU fine-tunes the Comprehensive Caption Composer ($C^3$) to summarize these three different captions into a comprehensive and accurate text caption. In the following, we will elaborate on the visual semantic extractor and the Comprehensive Caption Composer ($C^3$) in turn.

**Visual Semantic Extractor.** It first adopts an *image captioner* to generate a global and rudimentary description for an image, denoted by $\mathbf{T}_c$. Here any *image captioner* model is supported, such as OFA (Wang et al., 2022b) and BLIP (Li et al., 2022). In this work, LIU employs OFA (Wang et al., 2022b) because of its better performance and higher efficiency. For the same image, LIU uses the GRIT model (Wu et al., 2022) as its *dense captioner* to propose regions and also to generate a description, denoted by $\mathbf{T}_d$. Each description has several bounding box captions, describing the content of each box with coordinates. Thus, LIU generates fine-grained captions. Finally, for the *region-level captioner*, LIU implements it by the "segment anything" model (Kirillov et al., 2023) and uses it to extract region-level captions. However, segment anything model presents an over-segmentation issue and includes an excessive number of objects. To address this issue, LIU adopts a semantic segment anything model (Chen et al., 2023) to obtain semantics for each region, retaining only the largest top-K regions. In this work, we follow (Chen et al., 2023) and set $K = 10$. For this region-level caption, we denote it as $\mathbf{T}_s$, formatted as region captions, and also follow region coordinates. Note we only need to generate these three captions of all training data before the training and do not repeatedly generate these captions in the training phase to improve efficiency.

**Comprehensive Caption Composer ($C^3$).** In this stage, we employ a large language model (LLM), also called $C^3$ in this work, to synthesize the extracted visual semantics captions into a coherent textual description. However, the generated image captions, dense bounding box captions, or region-level captions may not always be accurate, and the generated bounding box may have displacement with ground-truth position. To ease this issue, the $C^3$ model uses all captions, especially for the object coordinates in the dense captions and region-level captions, to reason and correct the object positions in the image and object relations. Meanwhile, the $C^3$ model also removes the repeated objects and their corresponding captions. Next, we introduce how to train our $C^3$ model to achieve these targets.

We adopt the widely LLAMA (Touvron et al., 2023) as our $C^3$ model and employ the instruction tuning method to train it via a Language Modeling loss. In this training process, for randomly

Table 1: **The data template to generate training data samples.** The auto-regressive large language model is trained to predict the masked text $\mathbf{T}_\mathrm{p}$.

```
System Prompt <STOP>
Input:Image Caption:Tc;Dense Caption:Td;Region Caption:Ts <STOP>
Output:Tp <STOP>
```

masked tokens, the model is trained to predict these masked tokens. Given an input sequence $X = x_1, x_2, \ldots, x_N$, where each $x_i$ is of the format as shown in Table 1, composed of a system prompt message, input, and output. For each sample $x_i$, its system prompt is to set a specific context for the GPT model's responses (see Appendix A.1). For $\mathbf{T}_\mathrm{c}$, $\mathbf{T}_\mathrm{d}$ and $\mathbf{T}_\mathrm{s}$, they are generated by our visual semantic extractor. Regarding $\mathbf{T}_\mathrm{p}$, we use GPT3.5-Turbo (OpenAI, 2023a) and GPT4 (OpenAI, 2023b) to generate. Specially, we first collect the captions $\mathbf{T}_\mathrm{c}$, $\mathbf{T}_\mathrm{d}$, $\mathbf{T}_\mathrm{s}$ of 6K COCO training images. Then we use GPT3.5-Turbo (OpenAI, 2023a) to generate the corresponding $\mathbf{T}_\mathrm{p}$ of 5K COCO training images, and adopt GPT4 (OpenAI, 2023b) to generate 1K higher-qualified $\mathbf{T}_\mathrm{p}$.

Next, we can train LLAMA to predict the comprehensive caption $\mathbf{T}_\mathrm{p}$ based on the three caption inputs $\mathbf{T}_\mathrm{c}$, $\mathbf{T}_\mathrm{d}$ and $\mathbf{T}_\mathrm{s}$. In this way, our training loss can be formulated as

$$L_\mathrm{S} = -\frac{1}{|\mathbf{T}_\mathrm{p}|} \sum\nolimits_{y_i \in \mathbf{T}_\mathrm{p}} \log P(y_i | \mathbf{T}_\mathrm{c}, \mathbf{T}_\mathrm{d}, \mathbf{T}_\mathrm{s}). \quad (1)$$

But training LLAMA is expensive, especially for GPU memory, due to its large model. So we use a Parameter-Efficient Fine-Tuning method, LoRA (Hu et al., 2021), to adapt the attention weights.

In the experiment, we also observe that while the trained LLAMA excels at summarizing captions, it grapples with accurately understanding positional information of objects. To enhance positional reasoning, we introduce two *position-guided auxiliary tasks* with 1K samples each from COCO ground-truth object annotations. Specifically, the first task aims to predict the position of a bounding box based on its coordinates in the image, while the second task predicts the relative position between two bounding boxes. Please see more details in Appendix A.1. During the inference phase, given $\mathbf{T}_\mathrm{c}$, $\mathbf{T}_\mathrm{d}$, and $\mathbf{T}_\mathrm{s}$, the model is to predict the caption $\mathbf{T}_\mathrm{p}$. Notably, this caption is computed once and stored on the hard disk for subsequent reuse.

## 2.3 MODEL EVALUATION

In this work, we mainly evaluate our LIU on Dataset Refinement and Vision-language Task.

**Dataset Refinement.** Large-scale image-text pre-training corpora often suffer from a significant misalignment, where the provided captions are either sparse or inaccurate. This discrepancy results in the "one-to-many" issue, where each image may correspond to multiple distinct texts within the corpus. While an image can possess various facets and interpretations, traditional image caption models may not encapsulate the entirety of the image's content or context. To ease this issue, we substitute the original image-text pairs in the dataset with our generated image-caption pairs.

**Vision-language Task.** We initially assess LIU on the popular image-text retrieval task. As shown in Figure 1, we feed the generated text caption and the original source caption into a frozen language model, e.g., BERT (Devlin et al., 2019) or Flat-T5 (Chung et al., 2022). We then utilize pairwise cosine similarity for the evaluation. Other vision-language tasks are reported in Appendix A.2.

## 3 EXPERIMENTS

For $C^3$ model, we use LLaMA-7B (Touvron et al., 2023). For the language model for vision-language tasks, we initially utilized the conventional Bert (Devlin et al., 2019) model in our experiments. To further explore the improvement brought by the large language model, we progressively replace BERT with public-available larger models, utilizing GPT-2 (Brown et al., 2020b), OPT (Zhang et al., 2022), and Flan-T5 (Chung et al., 2022). We train our trainable $C^3$ for 6 epochs with a batch size of 128 and AdamW (Loshchilov & Hutter, 2017) optimizer with a weight decay of 0.05 on 1 NVIDIA A100 GPU. During training, we apply a learning rate warm-up to 3e-4 and a linear decay with a rate of 0.85. For the dataset refinement tasks, we use 8 NVIDIA A100 GPUs. Due

Table 2: **Comparison of our work with related studies on the COCO retrieval dataset using our generated captions**. The gray line represents the results of a zero-shot retrieval evaluation. We compare all methods under the same overall training cost as explained in Sec. 3.1.1.

| Method | Text Modality | Token Len | COCO (5K test set) | | | | | | |
| | | | Image→ Text | | | Text→ Image | | | |
| | | | R@1 | R@5 | R@10 | R@1 | R@5 | R@10 | Average |
|---|---|---|---|---|---|---|---|---|---|
| BLIP (Li et al., 2022) | Original Caption | 13 | 69.4 | 90.5 | 95.6 | 53.8 | 79.4 | 87.1 | 79.3 |
| | LIU | 41 | 71.0 | 91.4 | 95.5 | 54.7 | 81.2 | 88.9 | 80.5$_{+1.2}$ |
| | Original caption | 13 | 41.5 | 66.2 | 76.4 | 30.8 | 55.4 | 65.9 | 56.0 |
| | LIU | 41 | 43.2 | 70.1 | 79.7 | 33.7 | 57.6 | 66.7 | 58.5$_{+2.5}$ |
| CLIP (Radford et al., 2021) | Original Caption | 13 | 60.1 | 84.9 | 92.8 | 48.3 | 75.0 | 84.4 | 74.3 |
| | LIU | 41 | 64.2 | 87.2 | 95.1 | 54.3 | 79.5 | 87.8 | 78.0$_{+2.7}$ |
| | Original Caption | 13 | 36.4 | 64.0 | 78.5 | 29.4 | 54.3 | 65.2 | 54.6 |
| | LIU | 41 | 41.3 | 64.0 | 80.4 | 32.5 | 58.7 | 68.5 | 57.6$_{+3.0}$ |
| ViLT (Kim et al., 2021) | Original Caption | 13 | 64.9 | 88.3 | 93.1 | 51.1 | 78.1 | 86.4 | 77.0 |
| | LIU | 41 | 66.2 | 88.7 | 94.2 | 53.6 | 79.5 | 87.9 | 78.4$_{+1.4}$ |
| | Original Caption | 13 | 38.4 | 67.7 | 77.2 | 30.2 | 52.4 | 65.3 | 55.2 |
| | LIU | 41 | 42.8 | 70.8 | 80.7 | 32.4 | 57.1 | 65.9 | 58.3$_{+3.1}$ |

Table 3: **Comparison with BLIP model pre-trained on different data sources** for VQA, NLVR$^2$, RefCOCO+ and COCO Captioning. ViLT and CLIP architectures cannot be evaluated on some tasks due to structure limitations. The average token length is also listed for comparison. We compare all methods under the same overall training cost as explained in Sec. 3.1.1.

| Dataset | Token Len | VQA | | NLVR$^2$ | | RefCOCO+ | | | COCO Caption | |
| | | test-dev | test-std | dev | test-P | val | testA | testB | B@4 | CIDEr |
|---|---|---|---|---|---|---|---|---|---|---|
| *CC3M* (Sharma et al., 2018) | 13 | 71.5 | 71.8 | 76.0 | 76.2 | 72.4 | 76.1 | 65.3 | 36.8 | 121.6 |
| *LIU*-CC3M | 47 | 73.5 | 74.4 | 78.0 | 78.4 | 75.5 | 78.9 | 68.7 | 37.6 | 127.2 |
| *YFCC15M* (Thomee et al., 2016) | 11 | 70.5 | 70.8 | 74.2 | 74.5 | 70.6 | 74.2 | 63.1 | 35.9 | 118.4 |
| *LIU*-YFCC15M | 53 | 72.1 | 72.7 | 77.4 | 77.9 | 73.5 | 76.8 | 66.4 | 37.2 | 122.5 |

to space limitations, we defer some experiments into the Appendix, including VQA, Captioning, and dataset refinement on Flickr30K.

## 3.1 DATASET REFINEMENT

### 3.1.1 ENHANCING IMAGE-TEXT RETRIEVAL DATASET WITH IMPROVED CAPTIONS

Addressing semantic ambiguity in image-text retrieval is pivotal, given that a single image can have multiple valid descriptions based on the observer's perspective or intent. Conventional image-text retrieval datasets often comprise short captions, making it challenging for models to retrieve accurate pairs, particularly as the data scale expands. By generating more detailed captions, we can provide a more comprehensive understanding of the image, catering to diverse use cases or user preferences.

We conduct comparative analysis via three representative vision language pre-training frameworks: ViLT (Kim et al., 2021), BLIP (Li et al., 2022), and CLIP (Radford et al., 2021). Within large-scale pretraining scenarios, we adopt the previous approach (Kim et al., 2021; Li et al., 2022) to train our model, i.e., pretraining on a 4M corpus (Wang et al., 2022a; Li et al., 2020) and then fine-tuning on the COCO retrieval dataset.

Table 2 reports the results we observe: **1**) In image captioning, there's a challenge known as the 'one-to-many' problem. Essentially, a single image can be described in various ways using short captions. Longer and more accurate captions will ease such problems. Take note that the average token length for LIU is 41, and is nearly three times the length 13 of the original caption. However, **our LIU and other methods share the same overall training cost**, since all methods use the same maximum token length during the tokenizer preprocessing stage. In the context of retrieval tasks, which are typically viewed as one-to-one evaluations, a better performance signifies more accurate matching. Our findings reveal that by generating more detailed captions, accuracy is enhanced across all architectures. This suggests an effective mitigation of the one-to-many problem.

**2**) Interestingly, the relative improvement on CLIP is more significant. The potential reason is that 1) CLIP is solely trained using contrastive loss (Hadsell et al., 2006), and may not well align image and caption, while our LIU can help alignment and thus has substantial improvements.

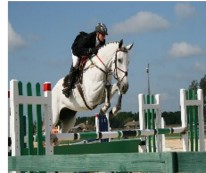

**Source Image**

**LIU wo Auxiliary Task:** ITM Score: **0.83**

**LIU:** ITM Score: **0.85**

There is a person jumping a white horse over an obstacle. The white horse is jumping over a green wooden fence that stretches from the left edge to the right edge of the image and is located towards the bottom of the image. There are two fences visible in the image - the green wooden fence and the green and white fence. The green fence is further towards the back of the image, and it is rectangular in shape.

A person jumping a horse over an obstacle. The horse is white and is jumping over a green wooden fence, which is situated towards the bottom of the image. The man positioned slightly above the fence and towards the left edge of the image. There are two fences visible in the image: a green and white fence and a green wooden fence. The green and white fence is situated more towards the center of the image, while the green wooden fence is located closer to the bottom. Additionally, there is a mountain background from the left to right edge of this image , as well as a white cloud in the top center.

**Caption.** ITM Score: **0.04**

You think selling your best friend is easy? Think again.

Figure 3: **Assessing ITM Score Distribution and Caption Quality.** Illustration of a low Image-Text Matching (ITM) score sample from the CC3M dataset (Sharma et al., 2018). Our model, LIU, generates a more detailed and contextually relevant caption, thereby achieving a higher ITM score. Text positioned incorrectly is highlighted in red, while correctly positioned text is marked in green.

### 3.1.2 REFINE NOISY VISION-LANGUAGE PRE-TRAINING DATASET

**Evaluate on Vision-Language Tasks.** One intriguing application of our LIU is to enhance the noisy vision-language datasets via our generated captions. In this experiment, we simply substitute the original captions in the dataset with our newly generated ones. Here we select the representative CC3M dataset, and respectively train BLIP on the original CC3M and our LIU-CC3M. Similarly, as aforementioned in Sec. 3.1.1, we also keep their overall training cost the same for fair comparison.

Table 3 summarizes the results of the popular image-text tasks, including Visual Question Answering (VQA) (Antol et al., 2015), Natural Language Visual Reasoning (NLVR) (Suhr et al., 2018), RefCOCO+ (Yu et al., 2016), and COCO Caption (Lin et al., 2014). One can observe that the model trained on LIU-CC3M outperforms the one trained on CC3M across all tasks. For instance, the score on the COCO Caption task is improved from 121.6 to 123.8. These results show that our method can effectively improve the quality of existing datasets. Interestingly, the model trained on LIU-CC3M also performed well on the visual grounding task, RefCOCO+. This suggests that **incorporating position-sensitive nouns significantly aids in the learning of visual grounding tasks**.

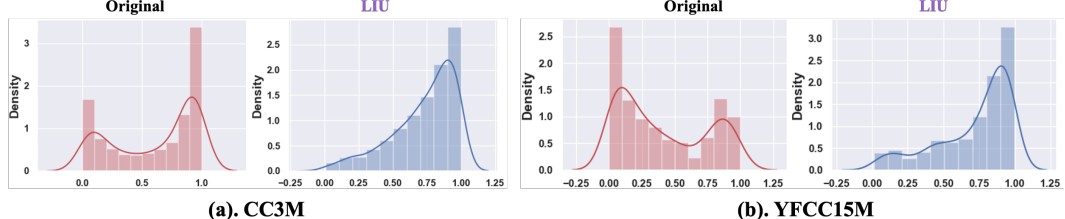

**(a). CC3M**                **(b). YFCC15M**

Figure 4: LIU **improves the Image-text Matching score** of two common used vision-language pre-training datasets significantly.

**Misalignment Cases.** Since large-scale vision-language datasets are often collected from the web, there are indeed substantial misalignment between the text and image pair, which is illustrated by Figure 3 by taking CC3M dataset (Sharma et al., 2018) as an example. This poses significant challenges for models to learn high-quality representations. Fortunately, the above improvement on many tasks shows LIU can generate a comprehensive caption and alleviate the misalignment issue, such as fixing the incorrect position descriptions in Figure 3.

**Alignment Analysis.** Here we evaluate the alignment between image and text pairs by computing the Image-Text Matching (ITM) score using a pre-trained BLIP model [1]. Figure 4 presents the ITM score distribution for both the original and our refined datasets. We observe a notable increase in the average ITM score for both the CC3M and YFCC datasets when using our generated captions compared to the original ones. Interestingly, while both vanilla datasets exhibit a peak around zero in their ITM distribution, our refined two datasets show a singular peak around one. This implies that our method can effectively alleviate the significant misalignment issues in these datasets.

---

[1]https://colab.research.google.com/github/salesforce/BLIP/blob/main/demo.ipynb

Table 4: **Comparison of our model with related works on the COCO retrieval dataset**. The zero-shot performance of LIU surpasses even that of the fine-tuned image-text retrieval models. Models that are inaccessible or can only be utilized through an API are grayed out. 'Train Param.' denotes the trainable parameters.

| Vision Modality | Train Param. | Time | COCO (5K test set) | | | | | | |
| | | | Image→ Text | | | Text→ Image | | | |
| | | | R@1 | R@5 | R@10 | R@1 | R@5 | R@10 | Average |
|---|---|---|---|---|---|---|---|---|---|
| Raw Image | 0M | 5m | 3.4 | 8.5 | 11.2 | 4.6 | 9.3 | 12.2 | 8.2 |
| OFA Caption (Wang et al., 2022b) | 0M | 5m | 49.6 | 73.5 | 82.1 | 36.1 | 62.9 | 73.3 | 62.9 |
| LIU | 0M | 5m | 51.4 | 75.1 | 82.8 | 38.1 | 64.3 | 76.2 | 64.7 |
| GPT4 Summared Text | 0M | 5m | 53.2 | 75.9 | 83.3 | 39.6 | 65.1 | 77.9 | 65.8 |
| Raw Image | 224M | 7h | 43.8 | 74.3 | 84.2 | 33.2 | 63.8 | 75.8 | 62.5 |

Table 5: **Comparing with related work on Flickr30K retrieval dataset**.

| Vision Modality | Train Param. | | Flickr30K (1K test set) | | | | | | |
| | | | Image→ Text | | | Text→ Image | | | |
| | | Time | R@1 | R@5 | R@10 | R@1 | R@5 | R@10 | Average |
|---|---|---|---|---|---|---|---|---|---|
| Raw Image | 0M | 5m | 6.3 | 11.0 | 15.7 | 6.6 | 10.2 | 14.5 | 10.7 |
| OFA Caption (Wang et al., 2022b) | 0M | 5m | 69.4 | 85.3 | 86.8 | 65.5 | 83.9 | 86.8 | 79.6 |
| LIU | 0M | 5m | 73.1 | 85.0 | 86.5 | 68.9 | 86.1 | 87.0 | 81.1 |
| GPT4 Summared Text | 0M | 5m | 74.5 | 86.3 | 88.8 | 70.5 | 87.4 | 88.5 | 82.7 |
| Raw Image | 224M | 7h | 73.3 | 87.1 | 89.2 | 67.6 | 85.1 | 87.4 | 81.7 |

## 3.2 IMAGE-TEXT RETRIEVAL

For image-text retrieval, we test on COCO (Lin et al., 2014) and Flickr30K (Plummer et al., 2015) under both zero-shot and fine-tuning settings. Here we use the BLIP (Li et al., 2022) model as a baseline and employ BLIP's text encoder (i.e., Bert (Devlin et al., 2019)) in our LIU to ensure fair comparison. Our approach is distinct in that it does not rely on image-text pair data, setting it apart from previous methods such as Flamingo (Alayrac et al., 2022), which are trained on large-scale datasets. Given the fundamental differences in data usage and the lightweight nature of our model, direct comparisons with these data-intensive and computationally heavy methods would not be meaningful or fair.

Table 4 and Table 5 report the evaluation results and show several observations. **1)** Raw Image method, with no learnable parameters, gives the poorest performance across all metrics, indicating the necessity of image-text pair training and the limitations of zero-shot retrieval without fine-tuning. **2)** When the learnable parameters increase to 230M, the retrieval performance improves dramatically, with substantial gains w.r.t. R@1, R@5, and R@10. **3)** Impressively, our *LIU model's zero-shot retrieval results sometimes surpass the Raw Image's fine-tuning results*, particularly w.r.t. R@1, e.g., improving from 43.8 to 51.4 on COCO. **4)** Our LIU is only inferior to GPT-4 which, however, uses a large amount of data and advanced algorithms for much longer training.

## 3.3 ABLATION STUDY

**Caption Model Selection.** Table 6a illustrates the impact of different image captioners to our method's performance. Both OFA and BLIP2 show superior performance compared to BLIP, and BLIP2 (Li et al., 2023) exhibits marginally better results than OFA. This shows that the quality and diversity of the image captions significantly affect the overall image caption generation. In this work, since BLIP2 needs much more GPU memory and computational time, we opt for the OFA model as the default choice to balance efficiency and performance.

**Language Model Selection.** Our LIU approach is compatible with a variety of language models of different scales and architectures. For an unbiased comparison, we use publicly accessible models and keep all other settings consistent. Table 6b reports the impact of different language model sizes

Table 6: **The ablation study of LIU**. We show the image-to-text retrieval result on COCO 5K.

| Method | Parameter | R@1 | R@5 | R@10 | | Method | Parameter | R@1 | R@5 | R@10 |
|---|---|---|---|---|---|---|---|---|---|---|
| BLIP (Li et al., 2022) | 224M | 37.5 | 63.8 | 75.4 | | BERTLM (Devlin et al., 2019) | 110M | 38.1 | 64.3 | 76.2 |
| OFA (Wang et al., 2022b) | 760M | 38.1 | 64.3 | 76.2 | | GPT-2 (Radford et al., 2019) | 1.5BM | 36.5 | 63.3 | 75.4 |
| BLIP2$_{OTP}$ (Li et al., 2023) | 3.1B | 39.0 | 67.0 | 77.4 | | OPT(Zhang et al., 2022) | 2.7B | 38.4 | 65.1 | 76.9 |
| BLIP2$_{T5XL}$ (Li et al., 2023) | 3.4B | 39.4 | 67.1 | 77.3 | | Flan-T5-XL (Chung et al., 2022) | 3B | 38.5 | 64.7 | 76.7 |

(a) The variation of Image Captioner.  (b) The variations of language model for retrieval.

Table 7: **Component ablation on COCO 5K retrieval dataset.**

| Image Captioner | Dense Captioner | Region Captioner | Image→Text | | | Text→Image | | | Average |
|---|---|---|---|---|---|---|---|---|---|
| | | | R1 | R5 | R10 | R1 | R5 | R10 | |
| ✓ | | | 49.6 | 73.5 | 82.1 | 36.1 | 62.9 | 73.3 | 62.9 |
| ✓ | ✓ | | 50.3 | **75.7** | 82.2 | 37.4 | 64.5 | 75.8 | 64.3 |
| ✓ | | ✓ | 50.7 | 74.6 | 81.5 | 37.1 | 64.2 | 75.5 | 63.9 |
| ✓ | ✓ | ✓ | **51.4** | 75.1 | **82.8** | **38.1** | **64.9** | **76.2** | **64.8** |

Table 8: **Ablation study on position-related auxiliary tasks.**

| Method | Auxiliary Tasks | RefCOCO | | | RefCOCO+ | | | $NLVR^2$ | |
|---|---|---|---|---|---|---|---|---|---|
| | | val | testA | testB | val | testA | testB | dev | test-P |
| baseline | | 71.4 | 75.3 | 63.4 | 72.4 | 76.1 | 65.3 | 76.0 | 76.2 |
| LIU | | $72.2_{+0.8}$ | $76.1_{+0.8}$ | $64.5_{+1.1}$ | $74.3_{+1.9}$ | $77.7_{+1.6}$ | $67.3_{+2.0}$ | $77.0_{+1.0}$ | $76.6_{+0.4}$ |
| LIU | ✓ | $73.5_{+2.1}$ | $77.2_{+2.9}$ | $65.1_{+1.7}$ | $75.5_{+3.1}$ | $78.9_{+2.8}$ | $68.7_{+3.4}$ | $78.1_{+2.1}$ | $77.9_{+1.7}$ |

(from 100M to 2B). As one might expect, larger models generally yield better results. However, we observe that improvements in retrieval performance plateau when the model size exceeds 1B. It's noteworthy that Flan-T5 and Flan-T5 XL, despite having the same architecture but different depths, do not provide substantial improvements. This may suggest that the retrieval task at hand may not necessitate extremely large models.

**Component Ablation.** Here we investigate the effects of the three components (Image Captioner, Dense Captioner and Region Captioner) in the vision semantic extractor in Table 7. One can observe that each captioner is effective. Image Captioner provides only a single overarching description, but frequently omits various cues and objects, while Dense Captioner and Region Captioner can effectively complement its description by providing a more fine-grained description.

**Impact of Auxiliary Positional-Guided Tasks.** Here we investigate the effects of auxiliary tasks on visual grounding ability. Specifically, we respectively train our LLaMA-based $C^3$ models within the LIU framework with and without auxiliary tasks. Specifically, we train them on the refined CC3M dataset generated by BLIP and then test them on visual grounding tasks. Here we choose RefCOCO, RefCOCO+, and $NLVR^2$ for evaluation as these tasks effectively test multimodal visual grounding abilities. Table 8 shows that LIU trained with auxiliary tasks outperforms the model trained without these tasks across all three visual grounding tasks, showing the effectiveness of our designed positional-guided tasks.

## 3.4 EXPLORATION AND DISCUSSION

**Image Generation from LIU Captions.** A fascinating aspect to explore is the potential of using a text-to-image model to regenerate an image from the caption generated by our method. We engage this possibility by utilizing the recently advanced Diffusion model (Ho et al., 2020). Specifically, we randomly selected some samples and generated informative captions with our method. Following this, we utilized the Stable-Diffusion (Rombach et al., 2022) and ControlNet (Zhang & Agrawala, 2023) models to regenerate images based on our captions. It is noteworthy that ControlNet incorporates the Canny Edge of the source image as a reference for a generation.

Figure 5 shows the regenerated images. It is intriguing to note that the most prominent objects are well-preserved in these images. We also attempt further visualizations using position-insensitive methods like Stable Diffusion (Rombach et al., 2022). As shown in Figure 3, the position of certain objects may sometimes be imprecise, and the color could potentially change. However, the captions generated by our method still align with the image far better than the original captions.

## 4 RELATED WORK

**Language Models in Vision-Language Tasks.** With the rise of robust pre-trained language models like BERT (Devlin et al., 2019) and GPT-3 (Brown et al., 2020a), their applications in vision-language tasks have become a research interest. Early models like (Lu et al., 2019; Li et al., 2019; Lin et al., 2021) utilized simpler models such as Bert for image textual descriptions, paving the way for integrated visual and linguistic understanding. Frozen (Tsimpoukelli et al., 2021) and TwT (Lin

| **Source Image** | **LIU result** | **T2I w Stable Diffusion** | **T2I w ControlNet** |
|---|---|---|---|
| 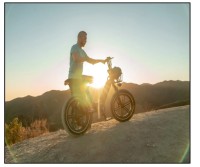 | In the image, three dogs can be seen sitting on a white rug. The first dog on the right is brown and is resting on a white blanket. The **second** dog is black and has a red collar around its neck and located on the left part. The third dog in the middle part is small and has black and brown fur. These dogs are sitting on a white background, while the squirrel is on the ground and has its tail extended. | 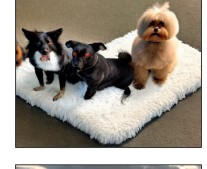 | 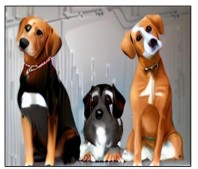 |
| | The image captured is of a man in the middle riding a bike on a sloping dirt road upside. The bike is black, and the man is dressed in green shorts. He is looking at the road. The scene is set outdoors under a blue sky. The front wheel of the bicycle is visible, alongside the wheel of a motorcycle. A bush lies on the side of the road. The sun setting behind him with a black mountain background. | 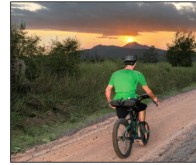 | 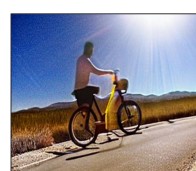 |

Figure 5: **Demonstrating Full-Cycle Image-to-Text-to-Image Transformation.** An example of transformation using the Stable Diffusion (Rombach et al., 2022) and stronger ControlNet (Zhang & Agrawala, 2023). The process begins with the transformation of the image into a paragraph using the LIU. Then, an image is regenerated from the derived paragraph, allowing for a direct comparison between the original and the regenerated images, thereby highlighting their key differences.

et al., 2022) employ a non-updatable BERT and a distinct visual encoder to harness pre-trained model advantages while cutting computational costs.

Large-scale Language Models (LLM) like OPT (Zhang et al., 2022), T5 (Raffel et al., 2020), and Flat-T5 (Chung et al., 2022) are now being adopted due to their enhanced representation capabilities. Such advancements aid in crafting superior vision-language models. BLIP2 (Li et al., 2023) incorporates frozen Flat-T5 and CLIP image encoder, training a Q-former for LLM adaptation.

In essence, the evolution in vision-language tasks has moved from basic models to sophisticated ones like GPT-3 and OPT. Their ongoing progress assures the development of more advanced models with better outcomes. Our work introduces a unique approach to leverage LLM for vision-language tasks.

**Extracting Semantics from Images.** Various strategies exist for semantic extraction from images. Object detection methods like Faster R-CNN (Ren et al., 2015) and YOLO (Redmon et al., 2016) pinpoint objects, whereas dense captioning (Johnson et al., 2016) describes them. Yet, these often miss contextual nuances and inter-object relations. Image captioning techniques, such as Show and Tell (Vinyals et al., 2015) and Show, Attend and Tell (Xu et al., 2015), offer single-sentence descriptions but overlook finer details. Visual Clues (Xie et al., 2022) adds object tags but omits their positions. SAM (Kirillov et al., 2023) introduces a robust segmentation model encompassing most real-world objects. Our method uniquely crafts a comprehensive caption for each image, emphasizing objects and their spatial arrangement. This offers a richer representation, capturing intricate details, relationships, and context, distinguishing it from conventional vision-language techniques.

## 5 CONCLUSION AND FUTURE DIRECTIONS

We introduced LIU, an innovative method aimed at streamlining the learning process for vision-language tasks. Our method effectively translates images into comprehensive textual descriptions, integrating both object identities and their positions. As an added bonus, LIU can also improve existing vision-language datasets, demonstrating significant efficacy in four downstream tasks.

Nonetheless, our approach is not without its limitations. It occasionally misses crucial information, such as color and object count, and can struggle with interpreting complex images featuring multiple objects. LIU may not be suitable for all vision-language tasks, particularly those requiring fine-grained visual grounding ability. Future efforts will aim to refine our technique, introducing more sophisticated and information-dense descriptions while also integrating nuanced visual cues. We further plan to extend the dataset refinement capabilities of LIU, which we believe holds the potential to significantly enhance the quality of vision-language tasks. We are hopeful that the strides made in this research will pave the way for future advancements in this rapidly evolving field.

## Reproducibility Statement

We have made significant efforts to ensure that our work is reproducible. The core code required to reproduce our findings is provided in the supplementary ZIP file. The primary components of this project are categorized into three sections:

Instruction Data Generation: Comprehensive guidelines and necessary scripts for data preparation are included.

Comprehensive Caption Composer Training ($C^3$): The implementation details and necessary files for training the $C^3$ model are provided.

Evaluation of Downstream Tasks: Instructions and code for evaluating on downstream tasks are included. Additionally, methods to use the $C^3$ model for refining the dataset and pre-training are detailed.

For the sake of reproducibility, we've removed all absolute paths from the code. To maintain anonymity in the review process, we have also redacted certain URLs from the code.

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

## A   Appendix

### A.1   Data Preparation

The supervision data used for fine-tuning LLAMA is 59K samples from Alpaca (Taori et al., 2023), the 6K generated instruction tuning samples, and 2K auxiliary samples, are both pure text format with three columns. Specifically, we randomly selected 6K samples from the COCO training dataset. Among these, the first 5K samples were processed with GPT-3.5, and the remaining 1K samples were processed with GPT-4. The 2K auxiliary samples come from the same COCO images. Each data entry consists of three parts: Instruction, Input, and Output. Then we concatenate these two TSV files directly and the shuffle is set to True during training. We introduce the instruction tuning data generation process next.

Table 9: **The system message to generate instruction data.**

```
Generate only an informative and nature paragraph based on the given .
information(a,b,c,d):
There are some rules:
1.  Show object, color and position.
2.  Use nouns rather than coordinates to show position information.
3.  No more than 7 sentences.
4.  Only use one paragraph.
5.  Describe position of each object.
6.  Do not appear number.
<STOP>
```

#### A.1.1   System Prompt

The system prompt is prepared for instruction-data generation. In practical, with pre-trained large language models (LLM) available, we need to add some prompt to guide the LLM. We show the system message in Table 9. Specifically, we give some rules to the Comprehensive Caption Composer ($C^3$) models. These rule prompt the model to generate native position-related paragraph.

To aid in understanding, we have formulated all this information into text and provided an illustrative example in Figure 6.

#### A.1.2   Instrocution Template of Auxilarity Task

To make the learning of position information simple, we first add two kinds of auxiliary tasks. Based on these tasks, we generate a lot of instruction samples first. Then we experimentally find the model learned with this auxiliary task very quickly and understand the position very well. The details are introduced next.

**First-order position relation.**   For this template, we simply describe the relation between the object and the position in each image. In practice, we split the image into nine grids first and then generate output annotation according to the coordinates of each object. The relation includes 9 text descriptions. For example, "top left" and "bottom right". We show the instruction data in Table. 10

Table 10: **The instruction tuning template for position-related auxilarity task 1.**

```
Predict where is the bounding box in the position of the image.
Image_size:  W × H
Bounding_box:  X1, Y1, X2, Y2
<STOP>
```

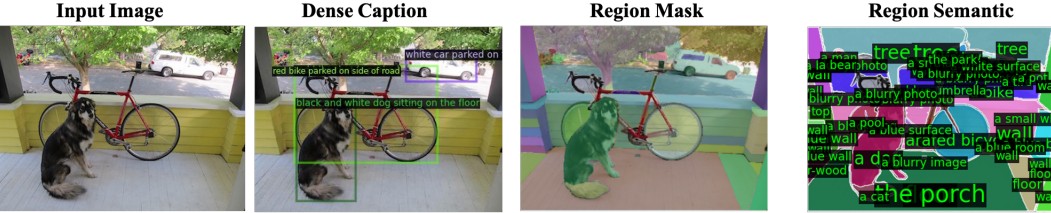

**(a). Region Semantic Visualization**

```
Question: Generate only an informative and nature paragraph based on the given information(a,b,c,d):

 a. Image Resolution:  369X276
 b. Image Caption: a dog sitting on a porch with a bike
 c. Dense Caption: black and white dog sitting on the floor: [64, 110, 155, 269]; red bike that is parked: [61, 59
, 283, 210]; white car parked on the street: [233, 37, 342, 84]; red motorcycle parked on side of road: [28, 35, 5
5, 61];
 d. Region Semantic: floor-wood: [0, 124, 374, 161]; floor-wood: [0, 188, 374, 97]; wall: [0, 105, 383, 106]; tree
: [21, 0, 359, 99]; tree: [20, 0, 225, 99]; the porch: [33, 59, 332, 53]; a dog: [67, 111, 85, 156]; a bicycle: [1
42, 65, 223, 47]; a bike: [191, 120, 91, 88]; tree: [242, 0, 139, 61];
 There are some rules:
        Show object, color and position.
        Use nouns rather than coordinates to show position information of each object.
        No more than 7 sentences.
        Only use one paragraph.
        Describe position of each object.
        Do not appear number.
```

**(b). System Prompt**

Figure 6: **System Prompt Generation.** In this example, we present an image from the COCO dataset, along with its corresponding region information, as illustrated in part (a). In part (b), we transform the object and bounding box information into text, demonstrating how visual elements can be systematically converted into a textual description.

**Second-order position relation.** For this task, we make this task harder and ask the model to learn the relation between bounding boxes. The instruction of second-order position relation between objects is given in Table. 11. For the output annotation, we use eight text annotations the same as first-order text annotation besides "middle".

Table 11: **The system message to generate instruction data.**

```
Predict the relation of BBOX1, using BBOX2 as reference.
BBOX1:  X1, Y1, X2, Y2
BBOX2:  X3, Y3, X4, Y4
<STOP>
```

## A.2 OTHER VISION-LANGUAGE TASKS

### A.2.1 IMAGE CAPTIONING

This task involves prompting the model to produce concise descriptions for each image. Given that we have already generated comprehensive image descriptions, this task can be straightforwardly assessed. We evaluate the task using two datasets: NoCaps(Agrawal et al., 2019) and COCO Captioning(Lin et al., 2014).

Specifically, we employ the BLIP(Li et al., 2022) model as a baseline for image captioning, trained on image-text pairs from the COCO Captioning dataset. As the NoCaps dataset does not include a training set, we assess a model trained on COCO Captioning for comparison. We prompt the language model to generate a brief summary based on the given sequences. Our method is evaluated in a zero-shot setting.

In line with previous methods (Wang et al., 2022a; Li et al., 2022), we requested the model to predict the caption autoregressively. We can effortlessly ask the LLM to generate a single sentence based on

Table 12: **The image captioning results on NoCaps and COCO Caption.** This task require a text decoder which is missed in ViLT and CLIP so we only report BLIP architecture. C: CIDEr, S: SPICE, B@4: BLEU@4.

| Method | Learn Param. | NoCaps validation | | | | | | | | COCO Caption Karpathy test | |
| | | in-domain | | near-domain | | out-domain | | Overall | | | |
| | | CIDEr | SPICE | CIDEr | SPICE | CIDEr | SPICE | CIDEr | SPICE | B@4 | CIDEr |
|---|---|---|---|---|---|---|---|---|---|---|---|
| Raw-Image | 361M | 33.1 | 8.2 | 26.1 | 7.8 | 18.3 | 6.1 | 25.5 | 6.5 | 17.7 | 46.5 |
| LIU $_{OPT350M}$ | 0M | 49.8 | 9.7 | 44.7 | 9.8 | 34.9 | 8.9 | 49.5 | 8.9 | 19.4 | 68.7 |
| LIU $_{OPT6.7B}$ | 0M | 53.4 | 10.3 | 48.3 | 10.4 | 37.5 | 9.4 | 53.7 | 9.4 | 22.3 | 75.3 |
| LIU $_{LLAMA7B}$ | 0M | 57.3 | 10.5 | 51.7 | 10.6 | 42.8 | 9.8 | 57.4 | 9.7 | 23.1 | 79.6 |

Table 13: **Comparing on Visual-Question Answering task**.

| Method | Learn Pararmeter | Time | VQA V2 | | OK-VQA |
| | | | val | test | test |
|---|---|---|---|---|---|
| Image | 0M | 7m | 1.3 | 2.1 | 1.4 |
| Image | 220M | 15h | 31.7 | 32.5 | 8.5 |
| LIU | 0M | 7m | 28.5 | 29.6 | 8.2 |

the provided sentence. Departing from the conventional prompts like 'an image of', we introduce a new prompt: **Summarize the paragraph into a single sentence for the image captioning task.** We experiment with various versions of LLM in this experiment and report the zero-shot evaluation results in Table 12. Despite nearly identical parameters (350M vs. 361M), our LIU outperforms models trained with image-text pairs for image captioning. This demonstrates the benefits of integrating large language models. We also observe that larger models generally yield better captioning results. At the same scale, the LLAMA(Touvron et al., 2023) performs marginally better than the OPT(Zhang et al., 2022) model. In conclusion, LIU proves highly effective for the captioning task. This success further substantiates the viability of our image-to-text approach.

### A.2.2 VISUAL QUESTION ANSWERING

Under our LIU framework, the evaluation of Visual Question Answering (VQA) becomes quite straightforward. We jointly input the question and generated captions, and then prompt the model to predict the masked parts directly. We employ a Transformer-based decoder model and evaluate its performance on the VQAv2 (Goyal et al., 2017) and Ok-VQA (Marino et al., 2019) datasets.

Specifically, each individual QA example is formatted as "Question: [question] Answer: [answer]" and concatenated. The results are displayed in Table 13. When compared to traditional models trained on image-text pairs, our zero-shot LIU performs considerably comparable. For instance, leads to 28.7 (vs 31.7) on VQA V2 val set but much less time.

### A.3 REFINEMENT RESULTS ON FLICKR30K

In this experiment, we refine the captions on Flickr30k using our generated captions. Similar to the caption refinement on the COCO dataset, we test both fine-tuning and zero-shot results across three representative frameworks.

The results are presented in Table 14. We note that LIU tends to perform better in most recall measures, particularly in R@1. This can partially be attributed to the model being trained solely with contrastive loss. We also observe that the language modeling loss of our generated captions is significantly higher than before (1.742 vs. 1.033 at the 20th epoch). This is a reasonable outcome given that the overall length is longer, making the prediction task more challenging.

As always, make sure to retain your LaTeX references in your actual document. For this context, I've substituted your LaTeX references with simple placeholders to enhance readability.

Table 14: **Comparison of our work with related studies on the Flickr30K retrieval dataset using our generated captions**. The gray line represents the results of a zero-shot retrieval evaluation.

| Method | Text Modality | Token Len | Flickr (1K test set) | | | | | | |
| | | | Image→ Text | | | Text→ Image | | | |
| | | | R@1 | R@5 | R@10 | R@1 | R@5 | R@10 | Average |
|---|---|---|---|---|---|---|---|---|---|
| BLIP (Li et al., 2022) | Original Caption | 11 | 79.3 | 93.5 | 96.5 | 59.4 | 81.9 | 88.0 | 83.1 |
| | LIU | 38 | 81.0 | 94.4 | 96.8 | 60.7 | 82.3 | 88.9 | $84.0_{+0.9}$ |
| | Original caption | 11 | 48.5 | 76.2 | 82.4 | 33.8 | 55.4 | 65.9 | 60.4 |
| | LIU | 38 | 51.2 | 77.3 | 82.9 | 36.7 | 57.6 | 66.7 | $62.1_{+1.7}$ |
| CLIP (Radford et al., 2021) | Original Caption | 11 | 70.3 | 86.4 | 94.1 | 52.5 | 77.3 | 86.2 | 77.8 |
| | LIU | 38 | 72.2 | 90.5 | 95.9 | 54.3 | 79.5 | 87.8 | $80.0_{+2.2}$ |
| | Original Caption | 11 | 45.3 | 73.2 | 80.3 | 30.9 | 54.3 | 65.1 | 58.2 |
| | LIU | 38 | 47.6 | 75.2 | 88.3 | 33.3 | 57.5 | 66.3 | $61.4_{+3.2}$ |
| ViLT (Kim et al., 2021) | Original Caption | 11 | 71.9 | 91.3 | 95.3 | 54.1 | 81.1 | 88.9 | 80.6 |
| | LIU | 38 | 73.4 | 91.7 | 95.6 | 56.8 | 81.8 | 89.7 | $81.5_{+0.9}$ |
| | Original Caption | 11 | 45.5 | 73.3 | 80.6 | 32.3 | 52.4 | 65.3 | 58.2 |
| | LIU | 38 | 46.8 | 70.8 | 80.7 | 35.2 | 56.2 | 65.8 | $59.3_{+1.1}$ |

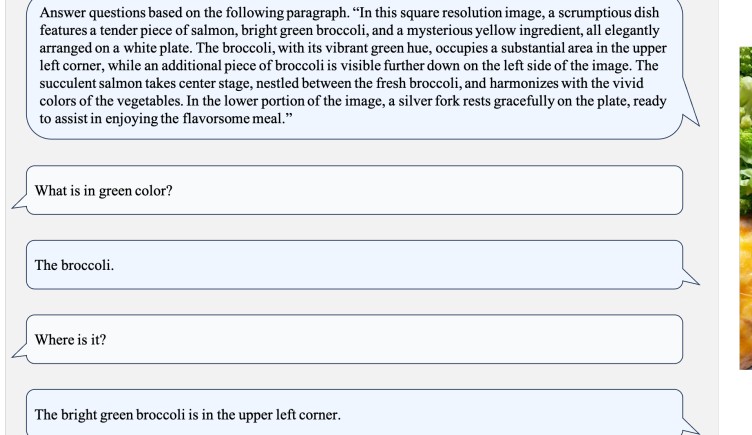

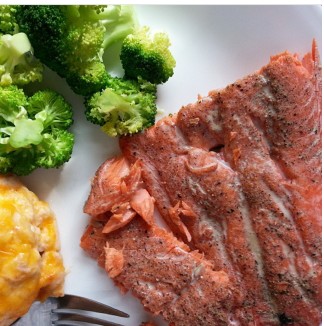

Figure 7: **Conversing with Images**: Leveraging the trained $C^3$ model enables straightforward interaction with images.

## A.4 FURTHER DISCUSSION

### A.4.1 CONVERSING WITH IMAGES

Upon training the Comprehensive Caption Composer ($C^3$) model, establishing a dialogue between images and text becomes fairly straightforward. We begin by introducing the system template ($System$). Then, based on this system template and user input, we generate a $Response$. More specifically, we directly input the caption and prompt the $C^3$ model to answer questions based on this generated caption.

The examples are displayed in Figure 7. It becomes apparent that the keys for color and position are easily interpretable. Once the detailed paragraph is obtained, users can also effortlessly interact with LangChain[2].

## A.5 FAILURE CASE

Despite the proficiency of our state-of-the-art semantic extractors and the thorough training of the Comprehensive Caption Composer ($C^3$) model, we still encounter occasional missteps. We've observed instances of significant misalignment between positions and images, and a tendency for the model to produce repetitive results.

---

[2]https://github.com/hwchase17/langchain

| Source Image | ILU | Generated Image |
|:---:|:---:|:---:|
| 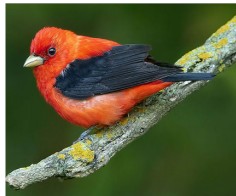 | In this image, we see a small bird perched on a branch of a tree. The wings of the bird have dark grey feathers, and its head is red with a **black beak** and eye. The bird is sitting on a branch that occupies most of the bottom half of the image. The branch has brown color, and there are **few leaves** visible on it. There is also a piece of a tree branch with yellow li li li li li li li li li li visible in the image. The flying bird is also red and can be seen in the top center part of the image. | 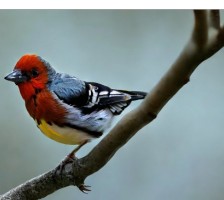 |

Figure 8: **Failure cases analysis**. We show wrongly predict, grammar typos and repeated words. We also observe the generated image with Stable Diffusion (Rombach et al., 2022) have looks different from source image.

Examples of these errors are demonstrated in Figure 8. We note that the trained $C^3$ model sometimes incorrectly synthesizes information, leading to a number of typographical errors. Text similarity measures may present a potential solution to mitigate the issue of repetition; we intend to explore this approach in future work.

### A.6 EXTENDED VISUALIZATION

This section presents additional samples that pose a challenge for the complete cycle of image-text and text-image transformation. To evaluate the model's versatility, we investigate three categories of data: indoor scenes, outdoor scenes, and egocentric scenes. The results are visually represented in Figure 9.

**Outdoor Scenes.** Outdoor scenes are common scenarios found in our data. These images usually feature prominent foreground objects and frequently include elements such as the sky and trees. Experimental findings show that the generated image often retains the semantic features of the original image. The images generated by ControlNet closely resemble the source images.

**Indoor Scenes.** Indoor scenes tend to be more complex due to the frequent presence of numerous objects. For instance, in the left image of Figure 9, a variety of furniture pieces can be observed in the bedroom, with some objects partially obscured. The generated image may miss some furniture, or variant in appearance.

**Egocentric Scenes.** We have chosen a sample from the Ego4D (Grauman et al., 2022) dataset for this category. We have found this to be particularly challenging due to the presence of irregular and open-world objects, some of which may never have been encountered before. This holds true for both the Diffusion and ControlNet models are failed to regenerate from the paragraph.

### A.7 OTHERS

Table 15: **Preserved Object Count**. We analyze the first 1000 samples from the COCO validation set, tallying the preserved objects within these images. The results are presented as a percentage, illustrating the ratio of the Top-K objects that are fully preserved in the generated captions.

| Object Tags | | | Object Position | | |
|:---:|:---:|:---:|:---:|:---:|:---:|
| Top1 | Top3 | Top5 | Top1 | Top3 | Top5 |
| 90.3 | 77.5 | 53.4 | 75.2 | 53.3 | 30.1 |

**Quantitative Experiments about Preserved Object Counts.** To demonstrate how much object information is preserved by the ILU, we conducted the following quantitative experiments:

Selection of Data: We chose the first 1000 samples from the COCO Validation dataset as our experimental base. Then, we use a Faster R-CNN model (ResNext152-C4) (Ren et al., 2015), trained

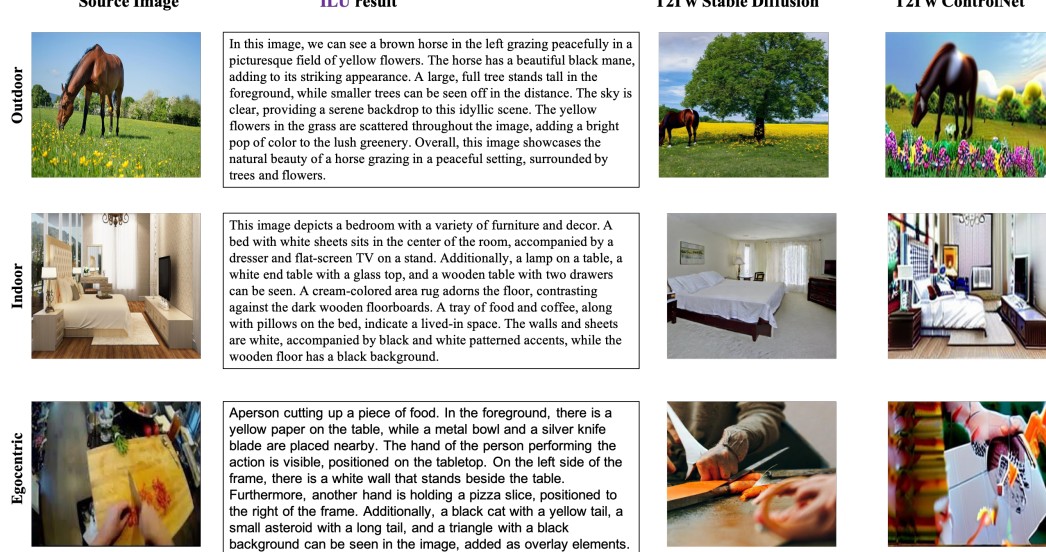

Figure 9: **The results of a comprehensive cycle of image-text and text-image transformations across various data domains.**

on 1600 classes from the Visual Genome to extract objects along with their tags and positions. At last, we selected the object tags corresponding to the top-5 highest confidence scores from unique classes. Object Tag Preservation: For the generated paragraph from the C model, we employed regular expressions to check if the detected object tags appeared in the paragraph. Position Preservation: Since the C model output deals with nouns rather than coordinates, computing an IoU score was challenging. Instead, we created a vocabulary of 28 commonly used location descriptors (e.g., "on the left of," "near," "in the upper-left corner"). If both the object tags and position nouns appeared in the same sentence, we marked the position as preserved.

The results of this experiment are displayed in Table 15. We find most of the object tags with the highest confidence scores are preserved. But the preserved position is only 53% for top3 objects. We guess the reason is partly from two issues with this process: Vocabulary Operation Noise: Our vocabulary-based approach introduced noise. Sometimes, the position noun, as in "A in B," was hard to define and missed in the current analysis. Synonymous Substitution: Occasionally, the generated text may have used synonyms for objects, such as substituting "computer" with "laptop." Therefore, it's important to note that the actual preservation of object tags and positions may be slightly higher than the reported metrics.

The models pretrained on LIU-generated data clearly show enhanced grounding ability on downstream position-related tasks. For the new ablation study about Quantitative Experiments on COCO Val: (a) Unique vs. Instance Object Class: The analysis reports on the unique object class rather than the instance object category. Take note that around 50% of samples in COCO have unique object classes less than or equal to 3, 74% have less than or equal to 5 (Lin et al., 2014), and the average category stands at 3.5. In other words, 50% of samples contain no more than 3 unique object classes. Although we utilize a 1600-class object detector to support a broader range of classes beyond COCO's predefined categories, most clear foreground objects should already be encompassed in COCO's statistical analysis. (b) Accuracy Analysis: The accuracy assessment considers only whether all predictions are correct rather than partial correctness. Meaning, if any object tags or positions are missing, it is marked as an error. Taking these details (a and b) into account, the upper bound for top-3 and top-5 accuracy should be significantly smaller than 100%. And it's clear that ILU can preserve object tags and position effectively.

**Definition of Second-Order Position Relations.** The concept of second-order position relations refers to the spatial relationships between objects within an image. This is complex, especially in images containing numerous objects, where overlaps and misplacements can occur.

Our approach to defining these relationships is quite straightforward: Utilizing Bounding Box Centers: Since the bounding boxes are square, we only consider the central point of each bounding box to determine the relationships between objects. Though this method is simple, it can indeed introduce noise into the analysis.

To handle overlaps, we experimented with excluding objects that had an overlap greater than 0.3 in terms of Intersection over Union (IoU). However, we found that this exclusion criterion resulted in the loss of many object pairs, and the impact on performance in retrieval tasks was negligible.

Table 16: **A simple data scaling experiment** on zero-shot retrieval text to image R@5 on COCO 5K test set.

| 1K GPT3.5 | 3K GPT3.5 | 5K GPT3.5 | 5K GPT3.5 + 1K GPT4 |
|-----------|-----------|-----------|---------------------|
| 31.5      | 32.3      | 32.4      | 33.7                |

**Data Scaling Effects and Investigation.** In this experiment, we present a comparison among models trained on 1K, 3K, 5K, and an additional 1K GPT4 samples. We report the Text to Image Recall 1 result on COCO5K retrieval task by replacing the original caption with our generated caption via $C^3$. Our observations reveal that utilizing 5K GPT3.5 data did not yield significant improvements over the model trained on 3K data. One primary reason for this outcome is that GPT3.5 tends to be somewhat weaker in reasoning, occasionally producing repeated objects and incorrect positioning. However, we recognize the potential for performance enhancement through the incorporation of more GPT4-generated data. Our findings thus lead us to conclude that the quality of the data plays a more pivotal role than the quantity in influencing retrieval performance.

