# OpenReview forum: "Linguistic Image Understanding"
_ICLR.cc/2024/Conference — ICLR 2024 Conference Withdrawn Submission_

### Official Review · Reviewer_kGwW · 2023-10-29

**Soundness:** 2 fair
**Presentation:** 2 fair
**Contribution:** 2 fair
**Rating:** 3
**Confidence:** 4

**Summary:**

This paper manuscript to directly transform an image into a text caption and then feed the generated captions into a frozen large language model for vision-language tasks. It employs OFA for image captioning, GRIT to generate dense bounding box captions, and (semantic) SAM to generate region-level captions. Following this, LLAMA is fine-tuned to merge the visual semantics captions into a coherent textual description. The experiments show that the image captions refined by LIU can effectively improve model performances on downstream tasks, such as COCO retrieval.

**Strengths:**

The experiments show that the image captions refined by LIU can effectively improve model performances on downstream tasks, such as COCO retrieval.

**Weaknesses:**

Novelty: Previous studies, such as BLIP which constructed BLIP-LAION-115M dataset, have already shown the benefits of refining image or video captions with captioner models. Recent advancements, exemplified by using LLAVA to generate dense captions in projects such as PixArt (for text-to-image generation), further highlight this trend.

Accuracy and Generalizability: The paper acknowledges potential inaccuracies in the visual semantic extractors used. Additionally, the fine-tuning of LLAMA is conducted on a relatively small dataset of 5K COCO images. Moreover, the training targets are generated using text-only GPT-3.5 and GPT-4, which could also introduce errors. These issues raise concerns about the robustness and generalizability of the model. The experiments presented in the manuscript do not sufficiently address these concerns. A more direct assessment of the caption quality generated by LIU, through human evaluation and comparison with LLAVA v1.5 and GPT-4V, is necessary to validate the claims made in the paper.

Incorporating OCR: The manuscript could be significantly enhanced by integrating Optical Character Recognition (OCR) capabilities, a critical feature in real-world applications, as demonstrated by GPT-4V.

In conclusion, the presented idea is not novel. Nevertheless, if LIU can be developed into an open-sourced image captioner capable of generating accurate and detailed descriptions for open-domain images—beyond the capabilities of models trained solely on 5K COCO images—it would represent a significant advancement.

**Questions:**

n/a

---

### Official Review · Reviewer_bm8R · 2023-10-30

**Soundness:** 2 fair
**Presentation:** 2 fair
**Contribution:** 2 fair
**Rating:** 3
**Confidence:** 4

**Summary:**

The paper proposes a framework called linguistic image understanding.  The framework consists of 3 pretrained image captioners to generate captions at different levels. Then a LLM is finetuned with the generated captions to provide promising performance. The supervision used during the finetuning is generated from GPT3.5/4.

**Strengths:**

Raise an interesting framework within LLM to improve the capabilities of image captioning.

The experiments show that the proposed method achieves better results in some tasks, like image-text retrieval and image captioning.

The paper is well-written, and the idea is easy to follow.

**Weaknesses:**

-- I think the idea of summarizing the captions of different levels with LLM is technically sound, while it is not clear the contribution of the proposed c3 module.  C3 module aims to distill the knowledge from GPTs by being finetuned with the supervision gained from them. So why not just use the GPTs as the c3 module or summarized the image? The knowledge distilled from the limited samples can bring bias, leading to a reduction in the performance of tasks. This is supported by Table 4. and Table 5. that the texts summarized by GPT4 have better performance than LIU for all metrics.

-- The experiment is not very comprehensive.

    -- Only BLIP is reported in Table 12. Although the performance seems good, it is not clear if the LIU is suitable for other kinds of captioning models or just BLIP. Please evaluate LIU with more different captioning models, so that we can know better about its capabilities and limitations.

    -- Similarly, VQA tasks also suffer some problems. Currently, there are many VQA works. The proposed prompt is only one option to achieve the question answering. Please try different prompt designs to show the efficiency. Here is some reference you may need:
    "From Images to Textual Prompts: Zero-shot Visual Question Answering with Frozen Large Language Models."
    "Prompting Large Language Models with Answer Heuristics for Knowledge-based Visual Question Answering"
    "An Empirical Study of GPT-3 for Few Shot Knowledge-Based VQA"

    -- The qualitative experiment in Figure 3. does not provide promising support. The negative selected in the example is too simple and obvious making it less meaningful as a comparison here. The captions including falsehood in different degrees can help to explain how good the LIU result is. Also, I think only one example is not quite sufficient to give a conclusion. It would be better to involve more examples.

    -- The experiment of T2I task lacks comparison. The generated images seem good, but no comparison is shown in the figures. You may use the original captions to generate the images with the same methods and compare them with the ones generated with LIU to show how better LIU can represent an image. Also, it would be more interesting if some quantitative evaluation metrics could be proposed to judge the performance of them.

**Questions:**

See the weakness

---

### Official Review · Reviewer_9a9c · 2023-11-01

**Soundness:** 3 good
**Presentation:** 3 good
**Contribution:** 2 fair
**Rating:** 5
**Confidence:** 4

**Summary:**

This paper proposes a simple text-centric approach to process vision-language tasks. The implementation of the entire pipeline is to convert images into comprehensive textual descriptions and input them into pre-trained LLMs. The experimental results confirm the feasibility of this method.

**Strengths:**

I. The proposed method is relatively simple to understand, and requires less annotated data.

II. The structure and logical expression of the article are clear。

**Weaknesses:**

The biggest constraint of this method is that  it appears to be an engineering integration work，especially the part of Vision Semantic Extractor.

**Questions:**

I. Some results of the experiment are not very convincing, especially Table 3 and Table 13 did not compare with the SOTA methods.

II. More discussion is needed on why both Dense and Region Captioners need to be retained simultaneously. From Table 7, it can be seen that the benefits of retaining both are not high, compared to the computational costs involved.

---

### Official Review · Reviewer_PcyU · 2023-11-02

**Soundness:** 2 fair
**Presentation:** 3 good
**Contribution:** 2 fair
**Rating:** 5
**Confidence:** 5

**Summary:**

This paper presents a text-centric vision-language framework called Linguistic Image Understanding. This work translates images into comprehensive textual descriptions, integrating both object identities and their positions. Additionally, the paper shows that the approach can be improved across multiple datasets.

**Strengths:**

1) Overall, this paper is well-written, and the technical details are easy to follow. I enjoyed reading this paper.

2) I find the idea of introducing improved structural knowledge using better textual descriptions appealing.

3) The results of the experiment strongly support the proposed approach, including the strong ablations that were performed.

**Weaknesses:**

**Technical Novelty.** The main contribution of this paper is the improvement of image captions through the use of three different types of caption knowledge: image captions, region captions, and dense captions. I believe that the general concept of adding structural knowledge to VL models is important, however, there has already been significant research in this area, which this paper ignores. In particular, [1] demonstrated that region descriptions generated from scene graphs that describe the image with objects, relations, and their attributes can improve compositional reasoning. Further, [3] demonstrated that adding denser and higher quality captions significantly improves these tasks. Finally, [4] demonstrates the importance of incorporating object localization into textual descriptions. It has already been suggested in all of these extensive works, which are not even discussed in the paper. Thus, I believe the novelty of the paper is therefore limited.


**Relation to Prior Work**. There are several works in that area that add structural knowledge using text, vision, and both for vision and language models, which the author should also add and discuss. In spite of the great importance of this topic, the related work paragraph at present has a limited scope, and there is no substantial discussion of approaches that already improves compositional reasoning and zeroshot performance. Here are some additional references the authors may wish to consider:

[1] Incorporating Structured Representations into Pretrained Vision & Language Models Using Scene Graphs. EMNLP 2023.

[2] When and why vision-language models behave like bags-of-words, and what to do about it. ICLR 2023.

[3] Dense and Aligned Captions Promote Compositional Reasoning in VL Models. NeurIPS 2023.

[4] Chameleon: Plug-and-Play Compositional Reasoning with Large Language Models

[5] Teaching Structured Vision & Language Concepts to Vision & Language Models. CVPR 2023.


**Experiments.** There are several benchmarks and tasks that are missing, and the authors could evaluate them as well. For example, Winoground, VL-checklist, SEED-Bench, SugarCrepe, etc. In addition, all the models used by the authors are outdated (CLIP, BLIP, and ViLT). It would have been appropriate to conduct these experiments with BLIPv2, LLaVa1.5, and more recent works. I do not believe that the results will be improved significantly over the most recent state-of-the-art results.

**Questions:**

I wrote above my concerns regarding the novelty.